# Impact of Melatonin Deficit on Emotional Status and Oxidative Stress-Induced Changes in Sphingomyelin and Cholesterol Level in Young Adult, Mature, and Aged Rats

**DOI:** 10.3390/ijms23052809

**Published:** 2022-03-04

**Authors:** Jana Tchekalarova, Zlatina Nenchovska, Lidia Kortenska, Veselina Uzunova, Irina Georgieva, Rumiana Tzoneva

**Affiliations:** 1Institute of Neurobiology, Bulgarian Academy of Sciences, Acad. G. Bonchev Street, Block 23, 1113 Sofia, Bulgaria; zuzania@abv.bg (Z.N.); lkortenska@abv.bg (L.K.); 2Institute of Biophysics and Biomedical Engineering, Bulgarian Academy of Sciences, Acad. G. Bonchev Street, Block 21, 1113 Sofia, Bulgaria; vesi.uzunova@abv.bg (V.U.); georgieva.irina5@gmail.com (I.G.); tzoneva@bio21.bas.bg (R.T.)

**Keywords:** age, melatonin deficit, behavior, oxidative stress, cholesterol, sphingomyelin, rat

## Abstract

The pineal gland regulates the aging process via the hormone melatonin. The present report aims to evaluate the effect of pinealectomy (pin) on behavioral and oxidative stress-induced alterations in cholesterol and sphingomyelin (SM) levels in young adult, mature and aging rats. Sham and pin rats aged 3, 14 and 18 months were tested in behavioral tests for motor activity, anxiety, and depression. The ELISA test explored oxidative stress parameters and SM in the hippocampus, while total cholesterol was measured in serum via a commercial autoanalyzer. Mature and aged sham rats showed low motor activity and increased anxiety compared to the youngest rats. Pinealectomy affected emotional responses, induced depressive-like behavior, and elevated cholesterol levels in the youngest rats. However, removal of the pineal gland enhanced oxidative stress by diminishing antioxidant capacity and increasing the MDA level, and decreased SM level in the hippocampus of 14-month-old rats. Our findings suggest that young adult rats are vulnerable to emotional disturbance and changes in cholesterol levels resulting from melatonin deficiency. In contrast, mature rats with pinealectomy are exposed to an oxidative stress-induced decrease in SM levels in the hippocampus.

## 1. Introduction

Aging is a natural phenomenon that involves many biological changes, including increased oxidative stress, DNA damage and diminished protein regulation, mitochondrial dysfunction, impaired immune responses, and vascular abnormalities [1]. Most of the aforementioned changes associated with aging are also valid for the pathogenesis of some neurodegenerative disorders, including Alzheimer’s disease, Parkinson’s disease, depression, and epilepsy. Therefore, the endogenous antioxidant system, which has decreased capacity with aging, might be directly involved in the pathological processes in the brain. The main effect of the hormone melatonin on antioxidant status in various tissues, other than direct radical scavenging, is related to its conversion into different antioxidant compounds, including cyclic 3-hydroxymelatonin, N1-acetyl-N2-formyl-5-methoxyquinuramin and N1-acetyl-5 methoxyquinuramin under certain conditions [2]. This gives us reason to consider melatonin as a broad-spectrum antioxidant with a much stronger potential than glutathione to neutralize free radicals and with a more effective protective effect on cell membranes than other antioxidants.

Changes in lipids and lipoproteins are associated with many diseases that affect the elderly. Sphingomyelin (SM) is a phosphosphingolipid involved in cellular membranes, including in the myelin sheath surrounding neuronal axons. It is metabolized to ceramide (Cer) in many cell types, as well as neurons due to oxidative stress [3]. The data in the literature support the suggestion that Cer accumulation might be related to a disturbance in the balance of oxidative stress homeostasis, thereby accelerating the aging process [4]. When SM conversion to Cer is enhanced, leading to accumulation of Cer, it provokes toxicity and apoptosis. Endogenous melatonin is directly involved in SM homeostasis [5,6], possibly via its suppression of the ASMase/Cer pathway in both the hippocampus of mice and cell lines [6]. 

Cholesterol, with its functional pleiotropism can play a major role in the regulation of neuronal functions and toxic cholesterol derivatives, which are formed with the participation of reactive oxygen species and become harmful factors for different neurodegenerative pathologies in the elderly [7]. Since the reports suggest that melatonin could be an effective cholesterol stabilizer in animal models of hypercholesterolemia [8], we can assume that melatonin, and its deficiency, might be a beneficial regulator of cholesterol concentration. 

One major disadvantage of most of the previous preclinical studies focused on the mechanisms underlying the aging process is related to the fact that younger rodents (i.e., <6 months of age) are used for this purpose. Summarizing the results of these age groups explains the translational failures in the development of drugs with the ability to slow down some aging-related processes. Therefore, it is necessary to understand the mechanism related to age-triggered or age-delayed changes in the brain to identify new therapeutic approaches to meet the needs of the elderly population as well as the development of multimodal health strategies that can contribute to their life quality and expectancy. The data support the idea that melatonin might be involved in some crucial mechanisms, including the impaired homeostasis of oxidative status in the CNS, and is responsible for accelerating or delaying the aging process at certain stages of ontogenesis [9,10,11,12]. Manipulation of the melatonin system during ontogenesis could directly impact the delicate balance between oxidants and antioxidants, which leads to a change in the concentration of some important lipids, and as a result, accelerates or slows down aging in the brain. In particular, controlling melatonin levels with aging could lead to the modulation of oxidative stress and concomitant biochemical changes, which is a rational strategy for preventing behavioral and biochemical disfunction of the elderly.

The present study aimed to test the hypothesis that melatonin deficit at different ages directly impacts the delicate balance between oxidants and antioxidants, leading to accelerated or delayed aging in the brain and associated behavioral and biochemical responses. Based on previous literature data regarding the crucial role of this hormone on some markers related to the aging process in experimental mice [9,10,11,12], specific age periods were chosen for testing. Clarifying the mechanisms associated with behavioral changes associated with aging, particularly the effect of melatonin deficiency on oxidative stress-induced changes in SM and cholesterol level, is essential for developing therapeutic strategies that can help improve the quality of life of the elderly.

## 2. Results

### 2.1. The Effect of Melatonin Deficit on Emotional Behavior in Young Adult, Mature and Old Rats

#### 2.1.1. Open Field (OF) Test

Aging was associated with reduced locomotor activity in the sham-operated rats (*p* < 0.001 and *p* = 0.016, 14 and 18-month-old rats, respectively, compared to 3-month-old rats) (Figure 1A, Appendix A). Pinealectomy induced age-dependent changes in the total distance traveled in the OF test with significantly elevated parameters detected in the youngest rats only (*p* = 0.029, 3-month-old pin rats compared to matched sham rats). Unlike ambulation, vertical activity (rearing) was not affected by age (*p* > 0.05) (Figure 1B, Appendix A). However, removing the pineal gland increased the frequency of rearing in young adult rats (*p* = 0.029, 3-month-old pin rats compared to matched sham rats). 

The anxiety-like behavior was exacerbated with aging, demonstrating that the sham-operated 3-month-old rats traveled more distance (*p* = 0.013 and *p* = 0.048, 14- and 18-month-old rats, respectively, compared to 3-month-old rats) and spent more time (*p* = 0.0031 and *p* = 0.026, 14- and 18-month-old rats, respectively, compared to 3-month-old rats) in the open arms, which is considered to be an aversive zone for rats (Figure 2A,B, Appendix A). The removal of the pineal gland attenuated anxiety in the youngest groups by increasing the distance (*p* = 0.02, 3-month-old pin rats compared to matched sham rats) and time (*p* = 0.002, 3-month-old pin rats compared to matched sham rats) in the open arms. However, pinealectomy did not affect anxiety levels in older rats (*p* > 0.05, 14- and 18-month-old pin rats compared to matched sham rats).

#### 2.1.2. Sucrose Preference Test (SPT)

The sham-operated 3-, 14- and 18-month-old rats showed no difference in preference to sweet solutions (*p* > 0.05) (Figure 3, Appendix A). Young adult and mature rats with pinealectomy developed anhedonia, indicating a depressive-like behavior, compared to their matched sham groups (*p* = 0.001) and (*p* = 0.021), respectively.

#### 2.1.3. Splash Test

Aging did not affect the duration of grooming behavior in the sham-operated 3-, 14- and 18-month-old rats (*p* > 0.05) (Figure 4, Appendix A). Both young adult and mature rats with pinealectomy showed deficits in self-caring behavior (*p* = 0.011) and (*p* = 0.023), respectively.

### 2.2. The Effect of Melatonin Deficit on Oxidative Stress Markers in the Hippocampus of Young Adult, Mature and Old Rats

#### 2.2.1. Superoxide Dismutase (SOD)

Regarding the oxidative stress profile as evaluated in the hippocampus, we detected age-related changes with a significantly elevated SOD activity in the 14-month-old sham rats compared to the matched 3-month-old rats (*p* = 0.028, Appendix A) (Figure 5A). Moreover, the pinealectomy suppressed the age-related elevation of the enzyme activity in mature rats (*p* = 0.011, 14-month-old pin rats compared to matched sham rats). The youngest and the oldest groups with pinealectomy showed diminished SOD activity compared to the matched groups (*p* > 0.05).

#### 2.2.2. Glutathione (GSH)

The sham groups (3-, 14-, and 18-month-old rats) showed no significant difference in the antioxidant GSH (Figure 5B and Appendix A). The 14-month-old rats with pinealectomy had decreased GSH levels in the hippocampus compared to the matched rats (*p* = 0.023). 

#### 2.2.3. Malondialdehyde (MDA)

Mature and old rats with pinealectomy showed elevated MDA in the hippocampus (*p* = 0.0055 and *p* = 0.012, 14- and 18-month-old sham rats, respectively, compared to matched 3-month-old rats) (Figure 5C and Appendix A ). Moreover, pinealectomy also elevated lipid peroxidation in mature rats compared to their matched group (*p* = 0.05). 

### 2.3. The Effect of Melatonin Deficit on Cholesterol Levels in Serum of Young Adult, Mature and Old Rats

As was expected, the 18-month-old sham rats demonstrated significantly elevated cholesterol levels in serum (*p* = 0.007 compared to 3-month-old rats) (Figure 6). Pinealectomy affected this marker of aging in the youngest group only (*p* = 0.012 compared to the matched sham group).

### 2.4. The Effect of Melatonin Deficit on Sphingomyelin Levels in the Hippocampus of Young Adult and Mature Rats

Pinealectomy did not affect behavioral responses, and oxidative stress parameters in the hippocampus of 18-month rats as was shown in Figure 1, Figure 2, Figure 3, Figure 4 and Figure 5. Therefore, the SM levels were measured in the hippocampus of sham and pin rats aged 3- and 14-months, respectively. Sham rats that were operated on at 14 months exhibited higher SM levels in the hippocampus (*p* = 0.032 compared to 3-month-old rats) (Figure 7). However, melatonin deficit attenuated SM levels in the hippocampus of mature rats (*p* = 0.003 compared to matched sham group). 

## 3. Discussion

In the present work, we demonstrated that while the aging process is accompanied by a gradual decline in motor activity, elevated anxiety and cholesterol levels in serum, and lipid peroxidation in the hippocampus, it does not produce depressive-like behavior in rats. Instead, our findings suggest that removal of the pineal gland induces age-specific changes in emotional responses, oxidative stress and SM levels in the hippocampus. Moreover, while the behavioral patterns related to emotional responses are more vulnerable to manipulation in young adult rats with pinealectomy, oxidative stress and SM levels in the hippocampus are dependent on melatonin deficit in mature rats. 

Motor activity is considered an important behavioral marker of aging. The detected age-associated decrease in motor activity in rats agrees with previous reports [13,14,15], confirming the assumption that this parameter reflects the change in functional state with aging. However, this behavioral marker is evident in a novel environment, suggesting that it is affected by stress levels. Indeed, Hofecker et al. (1979) [16] reported that home cage activity and locomotion in conditions associated with stress result in different effects during aging. The reported changes in motor activity with aging are probably related to the gradual decrease in the activity of the dopaminergic pathway in the striatum [13,17,18]. Moreover, in agreement with Sudakov et al. [19], the observed alteration in motor activity mainly affected the horizontal activity (ambulation) but not the vertical activity (rearing), which suggests that the exploratory behavior does not depend on aging. This agrees with our finding that aging does not produce depressive-like responses as measured in two tests, SPT and splash test. However, unlike exploratory and depressive-like behaviors, we found that two parameters in the OF tests expressing anxiety response were altered with aging. The detected decrease in distance and time spent in the aversive central zone of the OF might be due to low motor activity in rats aged 14 and 18 months and could represent a false-positive effect on anxiety. However, our results agree with other authors who reported that the gradual increase in anxiety level measured in the OF test, elevated plus-maze (EPM) test, and hole board test could be detected in mice and rats aged 4 to 24 months [19,20,21,22,23]. In contrast to our and other findings, Torras-Garcia et al. [24] reported that aging is characterized by a gradual decrease in anxiety, measured in the EPM test, in rats aged 3, and 17 to 24 months. However, Torras-Garcia et al. [24] did not confirm that aged rats exhibit lower anxiety levels in the OF test. Additionally, their aged group was subjected to food restriction from the age of 3 months, which might have affected their anxiety response differently than 3-month-old rats. 

Our results confirmed the previous report of Singhal et al. [22] suggesting that aging is not related to an increase in depressive-like responses in mice. However, other authors have reported opposite results for depressive patterns measured in the forced swimming test (FST) in female rats [23,24,25]. If the animals have decreased locomotion, depressive-like behavior measured in the FST could give false-positive results. Therefore, in the present study, we used the SPT and splash test to analyze the development of anhedonia or an increase in grooming as indicators of depressive responses. We also did not check this parameter in the FST because of the decreased motor activity in animals aged 14 and 18 months. 

In accordance with our previous work, in the present study, we confirmed that melatonin deficit related to removal of the pineal gland in young adult rats induced impulsive-like behavior in the OF test [26]. The rats with pinealectomy at three months showed hyperactivity and decreased anxiety. Moreover, depressive-like responses with anhedonia and diminished grooming in the youngest group with pinealectomy were verified in the SPT and splash test, respectively. We suggested that neurochemical and behavioral responses associated with melatonin deficiency at this age might model some aspect of melancholic depression in humans [26,27]. Our findings suggesting a lack of anxiety in the young adult rats with pinealectomy compared to the sham rats contradict other reports [28,29,30,31]. However, some crucial factors might explain the divergence in the reported effects of pinealectomy on anxiety behavior, such as age and time of testing after the removal of the pineal gland [31,32] and/or the hour of the day of examination [32], which determine the direction of the effects on anxiety. The depressive-like behavior associated with pinealectomy in adult rats agrees with our and other reports [26,29,33]. The disturbance in the ascending dopaminergic pathway from the midbrain to the nucleus accumbens (NA) is considered one of the mechanisms underlying the reluctance to consume sweet solutions (anhedonia) detected in the SPT [34]. Interestingly, Gaffori and Van Ree (1985) [35] reported that microinjected melatonin in the NA induced effects on motor activity and grooming behavior that were contrary to our findings with pinealectomy, suggesting that the underlying mechanism of melatonin effects is close related to the serotoninergic system in the NA. 

In the present study, we report for the first time that pinealectomy has no impact on anxiety and motor activity in a novel area in mature and old rats. In contrast, melatonin deficiency is crucial for these emotional responses in young adult rats. However, 14 months seems critical for the effects of the melatonin system on depressive behavior. Our previous studies reported that melatonin deficiency induced by either chronic light exposure or pinealectomy in young adult rats (3 months old) causes depression, including changes in motor activity and anxiety [26,36,37]. Exposure to a constant light regime also leads to characteristic disturbance in the sleep–wake cycle, sleep fragmentation, and changes in the duration and frequency of both NREM and REM sleep [38]. 

The findings that pinealectomy conducted in adult rats affects anxiety while removing the gland in adult and mature rats produce depressive responses, suggest that different mechanisms underlie these two neurobehavioral patterns and they deserve future exploration. In particular, the effect of pinealectomy induced at different stages of aging on the serotoninergic pathway in the NA may shed light on the mechanism involved. 

In agreement with the concept that a gradual disturbance in the homeostasis of endogenous antioxidants and free radicals underlies the aging process [39], sham-operated mature and aged rats had elevated lipid peroxidation, which was recently reported to correlate with memory decline in aged rats [40]. Our findings suggest that the impact of melatonin deficiency on the level of oxidative stress in the hippocampus is age-dependent. However, there is no direct link between the age-associated effect of pinealectomy on behavioral changes and oxidative stress. The limitation of the present study is that the markers of oxidative stress were measured at a one-time point during the light phase. Recently, we reported that while sham-operated young adult rats exhibited a circadian pattern of SOD, melatonin deficit induced by removal of the pineal gland, caused a flattened pattern of SOD and elevated lipid peroxidation, mostly during the dark phase when the metabolism of nocturnal rats is supposed to be higher and correlates with a surge in markers of oxidative stress [41]. Moreover, our previous and current reports suggest that the link between melatonin level and SOD is important in the dark phase in young adult rats. At the same time, in mature rats, the light period also seems crucial for the activity of this antioxidant enzyme. Melatonin is considered a powerful endogenous scavenger involved in the GSH cycle, particularly the reduced GSH [42]. Both endogenous and exogenous melatonin play an essential role in the production of GSH indirectly via increasing the activity of the antioxidant enzyme GPx [42,43]. The latter prevents the production of H_2_O_2_ and more reactive free radicals such as OH. In agreement with our previous report, removing the pineal gland did not affect the GSH level and lipid peroxidation in the hippocampus in young adult rats [41]. However, melatonin deficit exerted a phase-dependent effect on the MDA level at this age, with elevated lipid peroxidation in young adult rats during the active dark phase [41]. We reported that pinealectomy of mature rats attenuates the production of the antioxidant molecule GSH while exacerbating the lipid peroxidation in the hippocampus. The removal of the pineal gland does not affect GSH and MDA levels in the hippocampus in young adult and old rats, respectively. However, it seems that endogenous melatonin has a crucial role in mature rats for mitigation of free radicals’ production via promotion of GSH production. Moreover, oxidative stress-induced changes in the hippocampus resulted in reduced SM levels in the same structure. A previous study demonstrated that the mechanism underlying the anti-depressive activity of melatonin is related to suppression of Cer accumulation, possibly via inhibition of SM metabolism [6]. Altered lipid metabolism associated with Cer pathway metabolism of SMs in vulnerable to disturbed oxidative stress balance like the hippocampus can cause age- and disease-related synaptic dysfunction and neuronal degeneration [3,4]. 

## 4. Materials and Methods

The experiments in this study were performed in full agreement with the European Communities Council Directive 2010/63/EU for animal experiments. The research project was approved by the Bulgarian Food Safety Agency (# 300/№ 5888-0183).

### 4.1. Animals 

Male Wistar rats, aged 3, 14 and 18 months, were raised in the vivarium of the Institute of Neurobiology, Bulgarian Academy of Sciences. During the experimental procedures, all rats were kept in 3–4 rats per cage with food and water ad libitum and under standard conditions: temperature of 20 °C, 50–60% relative humidity and a 12 h light/dark cycle regime (on at 8:00 a.m.). 

### 4.2. Experimental Schedule and Surgical Procedures

A description of the steps in the experimental protocol is shown in Figure 8. The rats were evenly distributed in six groups: sham-operated and groups with pinealectomy (pin) aged 3, 14 and 18 months. The surgical procedure for the removal of the pineal gland was carried out as described in our previous studies [24,35,36]. Previously, we reported that removal of the pineal gland decreases the melatonin level in plasma, thereby leading to the abolition of the circadian pattern of hormonal release [41]. In brief, ketamine and xylazine were injected at a dose of 40 mg/kg (intraperitoneally, i.p.) and 20 mg/kg (intramuscular, i.m.), respectively, in the youngest groups. A dose of 60 mg/kg, i.p., and 30 mg/kg, i.m., respectively, was administered in rats aged 14- and 18-month-old. The fully anesthetized rats were fixed on a stereotaxic apparatus for a surgical procedure with pineal gland removal. In brief, after craniotomy (a small piece of the skull along with the suture lambda) by a dental drill, the gland, located beneath the venous sinus, was rapidly extracted via fine forceps. The procedure conducted on sham rats was the same except for pineal gland removal. Post-surgical action with an injection of antibiotics for three days was accomplished for all rats. 

### 4.3. Behavioral Tests

Behavioral tests were conducted one month after the surgery procedure.

#### 4.3.1. Open Field Test

The OF test was performed as previously [30]. The total distance traveled (cm) is a standard measure of general activity in the OF test. The length (cm) and time (s) spent in the aversive central area determine the anxiety level, while vertical movement (number of rearing events) shows exploratory behavior in rats. Except for the number of rearing events, which were detected manually, all parameters were measured automatically by a video tracking system (SMART PanLab software, Harvard Apparatus, Holliston, MA, USA). Each animal was tested individually and placed in the center of a polystyrene grey box (100 × 100 cm × 60 cm) for 5 min. 

#### 4.3.2. Sucrose Preference Test

The SPT explores the state of reluctance to consume sweet solutions (anhedonia), a behavioral marker of depression. The test was conducted according to our protocol described earlier in [36]. In the pretest, conducted for adaptation, each animal was placed in an individual cage with two identical bottles, which were filled with water up to 100 mL for 24 h, and then the water from one of the bottles was replaced with 1% sucrose solution for up to 3 days. The preference for sucrose consumption for 24 h (sucrose solution consumed (mL) ∗ 100/(water (mL) + sucrose solution (mL) was calculated in the test.

#### 4.3.3. Splash Test

Decreased stimulus-provoked grooming behavior (sec), which is considered an index of depressive-like behavior, was measured as described in our previous study [31]. In brief, an increase in grooming behavior was induced by the spraying of a 10% sucrose solution on the dorsal part of the body of the rat placed individually in the home cage. The total time spent grooming over a 5-min period was recorded immediately after applying the stimulus.

### 4.4. Oxidative Stress in the Hippocampus

#### 4.4.1. Superoxide Dismutase (SOD)

The tissue SOD activity was measured using a SOD kit (Cat No MBS162314, MyBioSource, Inc., San Diego, CA, USA). In brief, the rat hippocampal samples were weighed and homogenized in PBS (pH = 7.4) in the ratio of 300–500 mg tissue homogenized in 500 µL PBS, with the rotary homogenizer SONOPULS mini20 (BANDELIN electronic, Berlin, Germany). The samples were centrifuged for 20 min at 5000 pm, and the supernatants were carefully collected. Samples (40 μL) were mixed with 10 μL anti-SOD antibody, then 50 μL streptavidin-HRP was added to sample and standard wells. After incubation for 60 min at 37 °C, the plate was washed five times with wash buffer (1X). For color development, 50 μL of chromogenic solutions A and B were used. The addition of 50 μL of Stop Solution to all wells stopped the reaction and changed the color to yellow. SOD activity was measured by detecting the absorbance at 450 nm using a microplate reader (Tecan Infinite F200 PRO (Tecan Austria GmbH, Salzburg, Austria). The activity was calculated via 4PL regression and expressed as ng/mL.

#### 4.4.2. Glutathione

The GSH activity was estimated using a GSH kit according to the manufacturer’s instructions (Cat No MBS774706, MyBioSource, Inc., San Diego, CA, USA). A total of 10 μL (of each sample was prepared as in Section 4.4.1) was mixed in the designated 96-well plate with 40 μL sample diluent and 100 μL of HRP-conjugate reagent. Next, the plate was covered with a seal membrane, gently shaken and mixed for 60 min at 37 °C. After rinsing, 50 μL chromogenic solutions A and B were added to all wells for color development. The addition of 50 μL of Stop Solution to each well terminated the reaction and changed the color to yellow. The absorbance was measured at 450 nm using a microplate reader (Tecan Infinite F200 PRO (Tecan Austria GmbH, Salzburg, Austria), and the amount of GSH was calculated using 4PL regression. The GSH activity was expressed as mmol/L.

#### 4.4.3. Malondialdehyde

Lipid peroxidation was determined by measuring the MDA levels via an assay kit (Cat No MAK085, Sigma-Aldrich Co. LLC, St. Louis, MO, USA), according to the manufacturer’s protocol with few modifications. Briefly, the rat hippocampal samples were weighed and 25 mg of tissue were homogenized in 300 µL MDA Lysis buffer, containing 1xBHT (butylated hydroxytoluene) with a rotary homogenizer SONOPULS mini20 (BANDELIN electronic, Germany). The samples were centrifuged at 13,000× *g* for 10 min at 4 °C. Then 200 µL of the supernatant was transferred to a new tube, mixed with 600 µL TBA (thiobarbituric acid) and incubated for 60 min at 95 °C. After incubation, the samples were cooled in an ice bath for 10 min, centrifuged briefly, and 200 µL of the supernatant was transferred in duplicates to a 96-well plate. The MDA concentration was measured by detecting the absorbance at 532 nm using a microplate reader (Tecan Infinite F200 PRO (Tecan Austria GmbH, Salzburg, Austria), calculated based on a standard curve and expressed as nmol/mg. 

### 4.5. Sphyngomielin Level in the Hippocampus and Cholesterol Level in Plasma 

The SM activity was estimated using a SM kit according to the manufacturer’s instructions (Cat No. MBS265875, MyBioSource, Inc., San Diego, CA, USA). Briefly, the rat hippocampal samples were homogenized in concentration of 100 mg/mL in cell/tissue extraction buffer. After centrifugation at 4 °C and 10,000× *g* for 20 min, the supernatants were transferred to separate tubes and were heated for 5 min at 70 °C to become cloudy. After a second centrifugation at10,000× *g* for 2 min to remove the debris, 100 μL of each supernatant was transferred in duplicates to a 96-well plate. The SM concentration was measured by detecting the absorbance at 450 nm using a microplate reader (Tecan Infinite F200 PRO (Tecan Austria GmbH, Salzburg, Austria), calculated based on a standard curve and expressed as ng/mL. Serum cholesterol level was analyzed by an external certified laboratory (City Lab, Sofia, Bulgaria) using a commercial autoanalyzer. 

### 4.6. Statistical Analysis

Analysis of variance (two-way ANOVA) with the factors age (3-, 14- and 18-month-old rats) and surgery (sham and pinealectomy) followed by post hoc Bonferroni multiple comparison tests were used. The experimental data were presented as mean ± S.E.M. Statistical analysis was carried out by SigmaStat^®^ 11.0 (Systat Software, San Jose, CA, USA) and GraphPad Prism^®^6 (GraphPad Software, San Diego, CA, USA) software. The significant difference was accepted at *p* ≤ 0.05.

## 5. Conclusions

The results suggest that aging negatively affects some behavioral patterns associated with emotional responses via increased lipid peroxidation in the hippocampus and serum cholesterol. Furthermore, the impact of melatonin deficit on behavioral responses and oxidative stress is age-dependent. While removal of the pineal gland in young adult rats leads to impaired emotional responses and changes in serum cholesterol levels, mature rats with pinealectomy are vulnerable to an oxidative stress-induced decrease in the SM levels in the hippocampus. However, further exploration is needed to ascertain the precise impact of melatonin deficit on oxidative stress-induced sphingolipid metabolism directed to Cer accumulation and the neurodegenerative cascade of events associated with aging process.

## Figures and Tables

**Figure 1 ijms-23-02809-f001:**
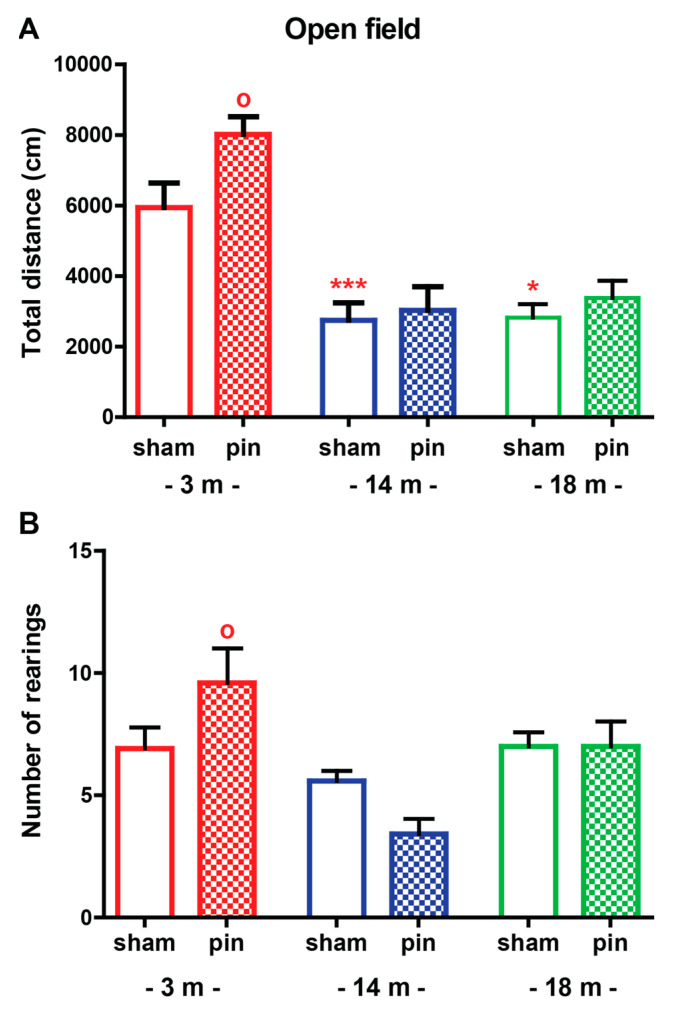
The effect of pinealectomy (pin) on total distance travelled (cm) (**A**) and number of rearing events (**B**) of 3, 14 and 18-month-old rats tested in the open field test. Data are presented as mean ± S.E.M. *** *p* < 0.001, 14-month-old sham compared to matched 3-month-old group; * *p* = 0.016, 18-month-old rats sham compared to matched 3-month-old group; ^o^
*p* = 0.029, 3-month-old pin compared to matched sham group (A); ^o^
*p* = 0.029, 3-month-old pin compared to matched sham group (B). 3-month-old rats—red; 14-month-old rats—blue; 18-month-old rats—green.

**Figure 2 ijms-23-02809-f002:**
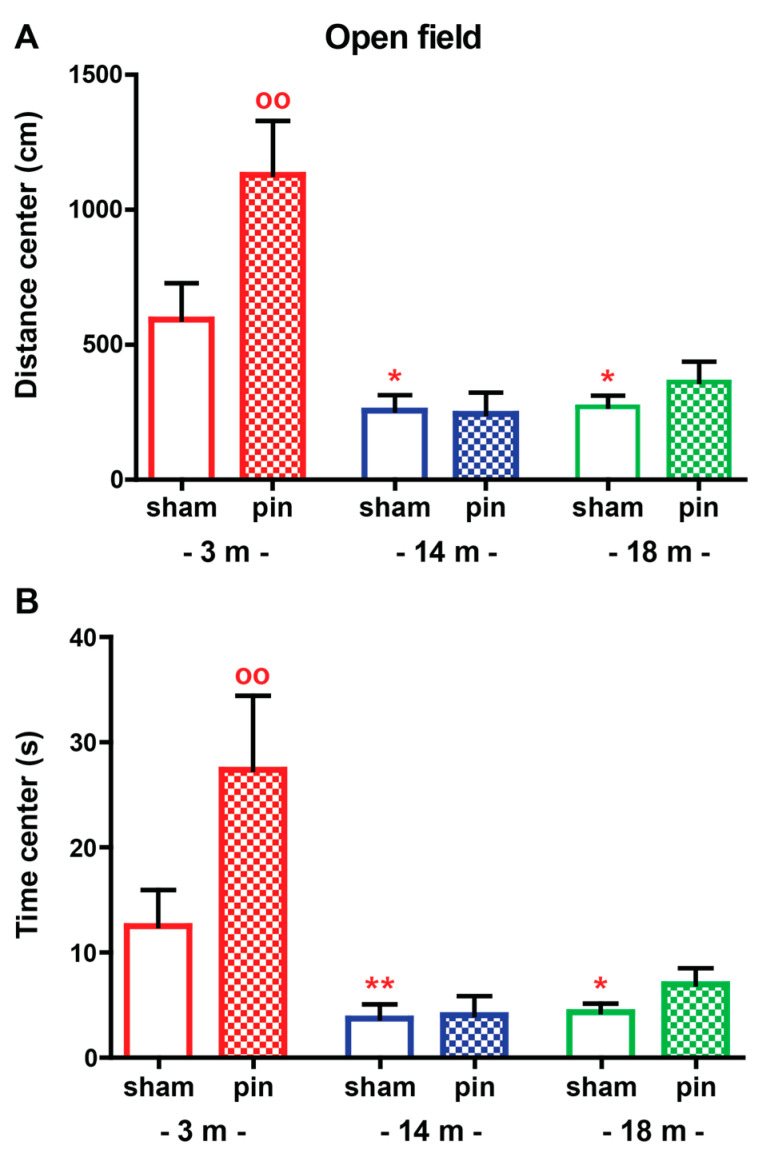
The effect of pin on distance traveled in the center (cm) (**A**) and time spent in the center (**B**) for 3-, 14- and 18-month-old rats tested in the open field test. Data are presented as mean ± S.E.M. * *p* = 0.013, * *p* = 0.048, 14- and 18-month-old rats, respectively, compared to 3-month-old rats; 3-month-old pin rats compared to matched sham rats (**A**); ** *p* = 0.003 and * *p* = 0.026, 14- and 18-month-old rats, respectively, compared to 3-month-old rats; ^oo^
*p* = 0.002 3-month-old pin compared to matched sham group (B). 3-month-old rats—red; 14-month-old rats—blue; 18-month-old rats—green.

**Figure 3 ijms-23-02809-f003:**
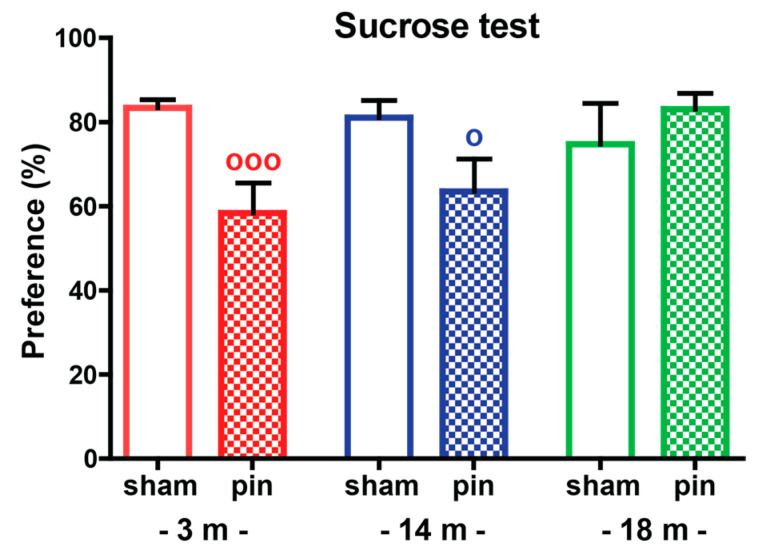
The effect of pin on preference to drink sweet solutions (anhedonia) of 3-, 14- and 18-month-old rats tested in the sucrose preference test. Data are presented as mean ± S.E.M. ^ooo^
*p* = 0.001, 3-month-old pin compared to matched sham group; ^o^
*p* = 0.021, 14-month-old pin compared to matched sham group. 3-month-old rats—red; 14-month-old rats—blue; 18-month-old rats—green.

**Figure 4 ijms-23-02809-f004:**
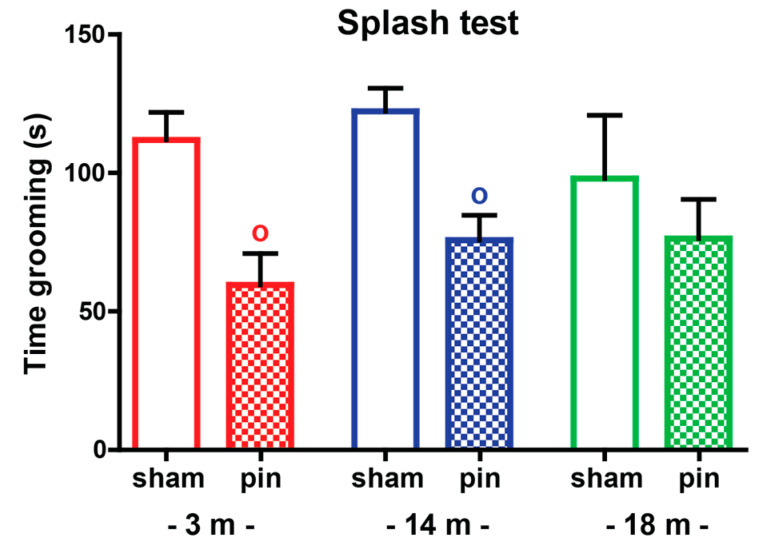
The effect of pin on grooming frequency of 3-, 14- and 18-month-old rats tested in the splash test. Data are presented as mean ± S.E.M. ^o^
*p* = 0.011, 3-month-old pin compared to matched sham group; ^o^
*p* = 0.023, 14-month-old pin compared to matched sham group. 3-month-old rats—red; 14-month-old rats—blue; 18-month-old rats—green.

**Figure 5 ijms-23-02809-f005:**
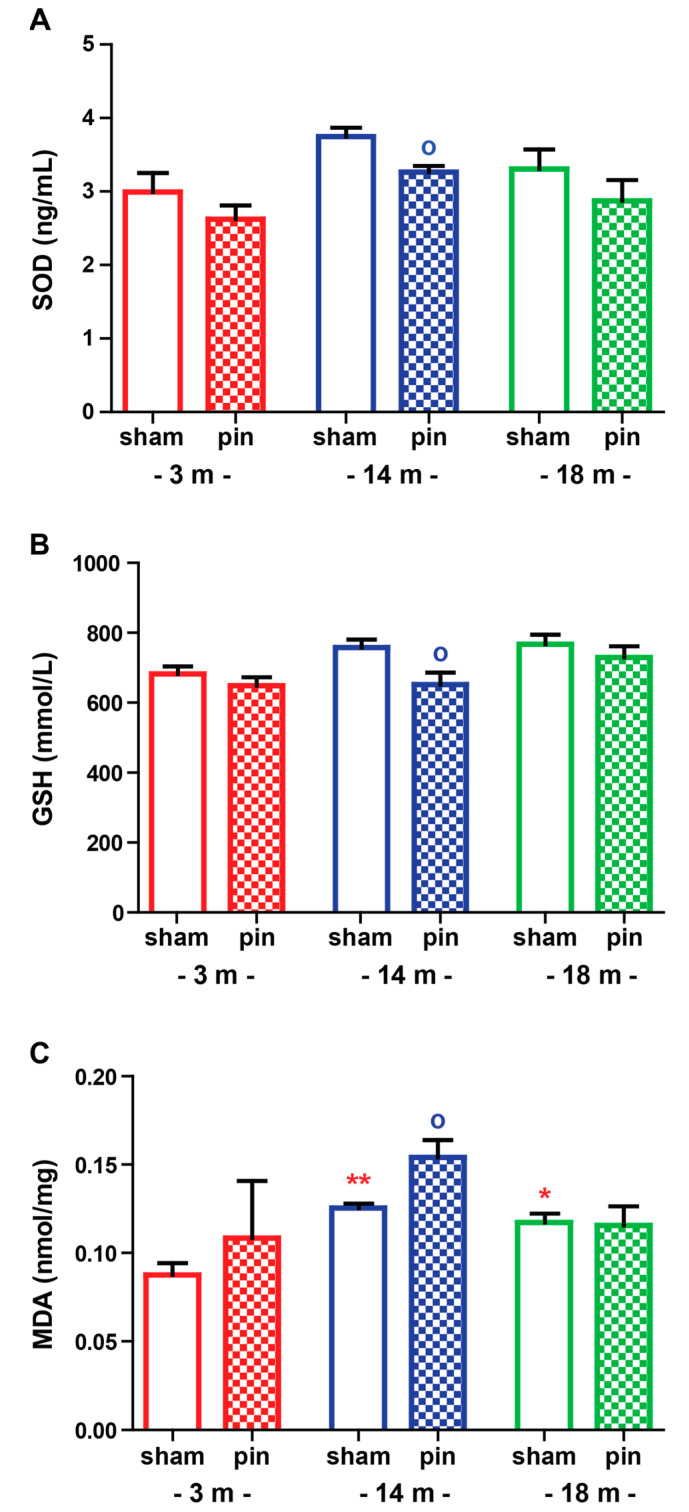
The effect of pin on SOD activity (**A**), GSH level (**B**) and MDA level (**C**) in the hippocampus of 3-, 14- and 18-month-old rats tested in ELISA. Data are presented as mean ± S.E.M. * *p* = 0.0284, 14-month-old sham compared to matched 3-month-old group; ^o^
*p* = 0.0108, 14-month-old pin compared to matched sham group (A); ^o^
*p* = 0.023, 14-month-old pin compared to matched sham group (B); ** *p* = 0.006, 14-month-old sham compared to matched 3-month-old group; * *p* = 0.012, 18-month-old sham compared to matched 3-month-old group; ^o^
*p* = 0.05, 14-month-old pin compared to matched sham group (C). 3-month-old rats—red; 14-month-old rats—blue; 18-month-old rats—green.

**Figure 6 ijms-23-02809-f006:**
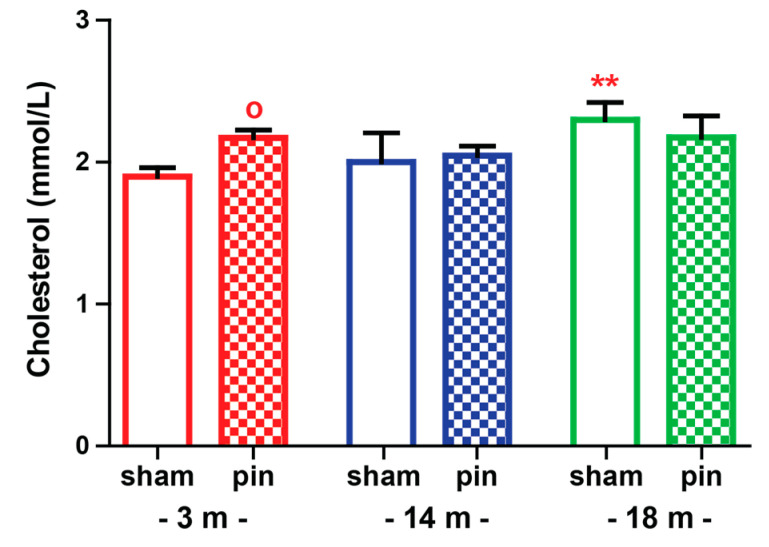
The effect of pin on cholesterol levels in serum of 3-, 14- and 18-month-old rats. Data are presented as mean ± S.E.M. ** *p* = 0.007 18-month-old sham rats compared to matched 3-month-old rats; ^o^
*p* = 0.0101, 14-month-old pin compared to matched sham group (A); ^o^
*p* = 0.023, 14-month-old pin compared to matched sham group (B); ** *p* = 0.0055, 14-month-old sham compared to matched 3-month-old group; 18-month-old sham compared to matched 3-month-old group; ^o^
*p* = 0.012, 3-month-old pin rats compared to matched sham group. 3-month-old rats—red; 14-month-old rats—blue; 18-month-old rats—green.

**Figure 7 ijms-23-02809-f007:**
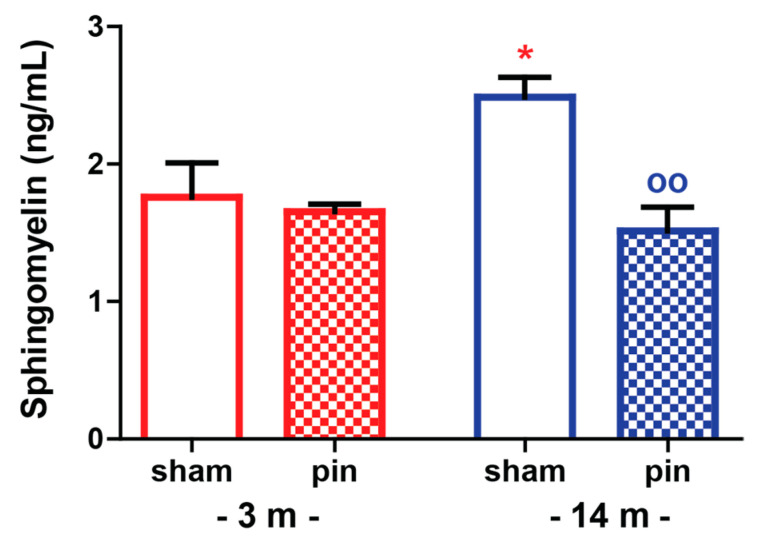
The effect of pin on sphingomyelin levels in the hippocampus of 3- and 14-month-old rats. Data are presented as mean ± S.E.M. * *p* = 0.032, 14-month sham compared to 3-month-old rats; ^oo^
*p* = 0.003, 14-month-old pin compared to matched sham group. 3-month-old rats—red; 14-month-old rats—blue; 18-month-old rats—green.

**Figure 8 ijms-23-02809-f008:**
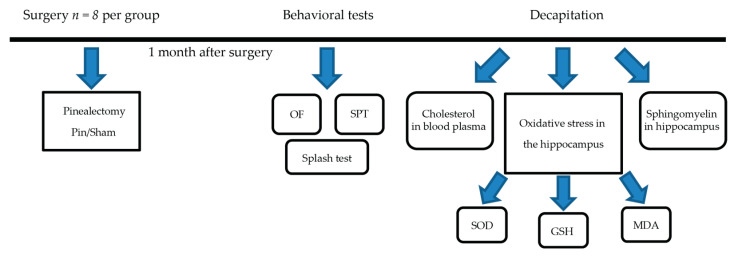
Experimental protocol with description of procedures and the number of animals used per group.

## Data Availability

Not applicable.

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
