# Peer review of "Impact of Melatonin Deficit on Emotional Status and Oxidative Stress-Induced Changes in Sphingomyelin and Cholesterol Level in Young Adult, Mature, and Aged Rats"

_ijms, 2022, doi:10.3390/ijms23052809_

Round 1

Reviewer 1 Report

Melatonin deficiency can cause various diseases, but it turns out that its excess can also harm us. Its production is lowest in the elderly. This hormone has a very strong effect, comparable even to steroids.

There is a relationship between failure in melatonin production in the pineal gland, an insufficient supply of this hormone to the body, and the occurrence of free radical etiology diseases such as neurodegenerative diseases, cardiovascular diseases, diabetes, cancer and others. Despite the development of molecular biology, numerous in vitro and in vivo studies, the exact mechanism of melatonin antioxidant activity is still unknown.

It has already been suggested that the loss of melatonin may cause the destruction of mitochondria by endogenous free radicals in certain regions of the brain that are especially important for memory and cognition. The loss of melatonin may be related to the loss of antioxidant protection in these parts of the brain.

The article refers the important for an aging population problem - how to slow down the processes associated with old age, and his medical implications cause  that it contains in the thematic profile of the periodical the International Journal of Molecular Sciences. The work has an experimental character, carrying in relatively new cognitive elements from the sphere of basic sciences. The author’s work opens important perspectives for measuring melatonin as a biomarker of early identification of neurodegenerative disorders.

In general, the manuscript is well written and the experimental quality and the conclusions drawn are adequate.

However, there are the major concerns:

1/ The autor’s findings suggest that removal of the pineal gland induces age-specific changes in emotional responses, oxidative stress and SM levels in the hippocampus. Melatonin is synthesized mainly by the pineal gland but it is also produced by the digestive system, blood cells, lens and retina of the eye, kidneys, thyroid, ovaries, cerebellum, bile, bone marrow and cerebrospinal fluid. In humans it is noteworthy that the concentration of melatonin in the gastrointestinal tract surpasses blood levels by 10-100 times and there is at least 400 times more melatonin in the gastrointestinal tract than in the pineal gland. In order to rule out the influence of extrapineal melatonin on the analyzed parameters, it would be worth determining its level in the blood of the tested animals.

2/ The authors wrote: “Manipulation of the melatonin system during ontogenesis could directly impact the delicate balance between oxidants and antioxidants…”. Indeed, this equilibrium  is very delicate, and its disturbance is influenced by many factors, including body weight. In addition, effects of melatonin on body mass and bone mass regulation have been reported. Melatonin is known for its role in energy expenditure and body mass regulation in mammals by preventing the increase in body fat with age - for example, a 10% weight loss will have a different effect on the health and welfare of a young growing and adult rat.  In the methodology of the described research, I did not find any information on the body weight of the animals. The compared groups of animals should be homogeneous in terms of this factor. Have they been?  

3/ The method of specifying p values should be standardized. It is enough to specify them with an accuracy of 3 decimal places. Entering 4 decimal places does not have much informational value.

Typographical errors:

  • “holesterol level” instead “cholesterol level”  (line 3 - in the title, 85)
  • “the regulaion of” instead “the regulation of” (line 54)
  • “neuronal functions ]” – remove ] (line 55)
  • “Since t reports” instead “Since the reports”? - (line 57)
  • “H2O2” instead “H2O2” (line 295)
  • “.OH” instead “OH” (line 295)

Author Response

We are thankful for your critical review and your valuable suggestions. Indeed corrections and modifications are helpful for the optimization of the manuscript. We hereby addressed your raised comments point-by-point basis on the followi.

Reviewers' comments:

Point # 1 The autor’s findings suggest that removal of the pineal gland induces age-specific changes in emotional responses, oxidative stress and SM levels in the hippocampus. Melatonin is synthesized mainly by the pineal gland but it is also produced by the digestive system, blood cells, lens and retina of the eye, kidneys, thyroid, ovaries, cerebellum, bile, bone marrow and cerebrospinal fluid. In humans it is noteworthy that the concentration of melatonin in the gastrointestinal tract surpasses blood levels by 10-100 times and there is at least 400 times more melatonin in the gastrointestinal tract than in the pineal gland. In order to rule out the influence of extrapineal melatonin on the analyzed parameters, it would be worth determining its level in the blood of the tested animals.

Response : We verified the model of melatonin deficit and reported in our previous study that while the sham-sed group showed a circadian pattern of plasma melatonin level peaked at dark and with a nadir at the light, the removal of the pineal gland entirely suppressed melatonin secretion at dark (Tchekalarova et al., 2020, https://doi.org/10.1016/j.neulet.2019.134637). We added a sentence in Methods (4.2. Experimental schedule and Surgery procedure) concerning the above-mentioned issue (356). line . The endocrine function of melatonin is closely associated with the pineal gland where it is released in the blood while in other tissues, including the gastrointestinal tract it behaves as a paracrine hormone. Previously, we reported that after removal of the pineal gland the circadian pattern of melatonin release is abolished showing a flattened 24-h pattern.

Point # 2 The authors wrote: “Manipulation of the melatonin system during ontogenesis could directly impact the delicate balance between oxidants and antioxidants…”. Indeed, this equilibrium  is very delicate, and its disturbance is influenced by many factors, including body weight. In addition, effects of melatonin on body mass and bone mass regulation have been reported. Melatonin is known for its role in energy expenditure and body mass regulation in mammals by preventing the increase in body fat with age - for example, a 10% weight loss will have a different effect on the health and welfare of a young growing and adult rat.  In the methodology of the described research, I did not find any information on the body weight of the animals. The compared groups of animals should be homogeneous in terms of this factor. Have they been?  

Response: We’re thankful for this relevant question. Indeed, another focus of our study related to the effect of melatonin deficit in aging is on different physiological and metabolic parameters, closely associated with the aging process, including energy metabolism, to specify their role responsible for delaying or accelerating aging in this condition. This study is submitted already and still under a review. The impact of aging on B.W. increase (the effect of melatonin deficit in different ages), food intake, VO2, VCO2, E.E., and RER, were explored by indirect Calorimetry. Sure the groups were homogenous in terms of this factor.

Point # 3 The method of specifying p values should be standardized. It is enough to specify them with an accuracy of 3 decimal places. Entering 4 decimal places does not have much informational value.

Response: We’re thankful for this remark. The issue was taken into consideration and corrected.

 Typographical errors:

All typos were corrected.

Reviewer 2 Report

The manuscript presents interesting and valuable work, which is within the scope of the Journal. I recommend the publication of the submitted manuscript in the International Journal of Molecular Sciences in the present form.

Author Response

We are thankful for your critical review and your valuable suggestions. Indeed corrections and modifications are helpful for the optimization of the manuscript.